# Comparison of quality of life between elderly and non-elderly adult residents in Okuma town, Japan, in a post-disaster setting

**Varsha Hande, Makiko Orita** *, **Hitomi Matsunaga** , **Yuya Kashiwazaki** ,
**Yasuyuki Taira, Noboru Takamura**

Department of Global Health, Medicine and Welfare, Atomic Bomb Disease Institute, Nagasaki University
Graduate School of Biomedical Sciences, Nagasaki, Japan

* orita@nagasaki-u.ac.jp

## Abstract

With the growing elderly population who are susceptible to poor health, improvement of their quality of life is essential. In the post-disaster setting of the 2011 Fukushima Dai-ichi nuclear power plant (FDNPP) accident, affected municipalities such as Okuma town commenced their recovery processes and lifted evacuation orders in 2019. This study examines the differences in self-reported mental and physical health status, social functioning, risk perception, and intention to return between elderly (age ≥65 years) and non-elderly (age 20–64 years) residents. Questionnaires were distributed to current residents and evacuees of Okuma. Results revealed that the elderly had a 1.4 times higher odds ratio (95%CI 1.0–1.8, p = 0.034) for having anxiety regarding radiation-related health effects on future generations and a 1.3 times higher odds ratio (95%CI 1.1–1.5, p = 0.001) for wanting to know about the release of FDNPP-treated water into the environment than the younger group. Elderly residents also demonstrated a 2.2 times higher odds ratio for reporting poor physical health than younger residents. Clearing misconceptions and disseminating coherent information will reduce risk perception among this group. Further in-depth research regarding the disposal of FDNPP-treated water and its perceived risks is required. Health promotion through the encouragement of social participation, improvement of surroundings to facilitate healthy behaviors, and enhanced access to health services will improve the quality of life of elderly Okuma residents.

## Introduction

As a result of the Great Japan Earthquake and tsunami on March 11, 2011, a nuclear accident occurred at the Fukushima Daiichi Nuclear Power Plant (FDNPP). Nuclear meltdowns occurred due to the loss of reactor core cooling, and hydrogen explosions took place in Units 1, 3, and 4 [1]. The disaster was characterized as a level 7 event (Major Accident) on the International Nuclear and Radiological Event Scale [2]. Consequently, radionuclides from the damaged plant spilled into the environment and an order to evacuate or remain indoors was issued

**Funding:** This work was supported as a Research Project on the Health Effects of Radiation, organized by the Ministry for the Environment, Japan. An Evaluation Committee was established for this project, which implemented appropriate research management by reviewing progress and implementation and providing advice when necessary. The progress and implementation for this study has also been checked. The funder was not involved in study design, data collection and analysis, or preparation of the manuscript.

**Competing interests:** The authors have declared that no competing interests exist.

to residents by the Prime Minister of Japan in his role as Director-General of the Nuclear Emergency Response Headquarters. At 20:50 that day, residents living within 2 km of the plant were ordered to evacuate. This order was extended to a 3 km radius, then to a 10 km radius, and finally, to a 20 km radius on March 12. These decisions resulted in a long-term and widespread evacuation of residents. As Okuma housed units 1, 2, 3, and 4 of the FDNPP, all 11,505 residents were forced to evacuate [3].

Based on an aggregation of knowledge (including data from Hiroshima and Nagasaki atomic bomb survivors) indicating an increase in cancer incidence and mortality associated with exposure to radiation doses higher than 100 millisieverts (mSv), the International Commission on Radiological Protection (ICRP) proposed the use of bands and reference levels of radiation according to exposure scenarios as one of its radiological protection strategies [4,5]. For the public under normal conditions, the ICRP (Publications 103, 111) recommends limiting annual exposure to radiation levels below 1 mSv (the reference first band of 1 mSv or less). The reference second band (1–20 mSv) applies to the protection of persons residing in contaminated areas. Actions should be taken to limit individual annual effective residual radiation levels towards the lower range of the band, with periodic reductions over time. Reference levels are adjusted based on relevant factors, such as contamination levels, decontamination progress, the sustainability of social, economic, and environmental life, and individual health status. The third band (20–100 mSv) applies to extreme cases, such as radiological emergencies. In July 2011, 4 months after the accident, the prefectural government measured radiation exposure doses in 4814 Okuma residents and found that 3374 (70.1%) received <1 mSv, 1284 (26.7%) received 1–2 mSv, 112 (2.3%) received 2–3 mSv, and 99.6% of residents were exposed to doses of 5 mSv or less [6]. International governing bodies such as the World Health Organization (WHO) and the United Nations Scientific Committee on the Effects of Atomic Radiation (UNSCEAR) have published evidence suggesting no increases in the risks of cancer or related disease incidence, either presently or in the future, as a result of this disaster [7,8]. Towns that lifted their evacuation orders early (2012), such as Kawauchi village, witnessed a return of almost 80% of their residents, but the same was not true for towns closer to the power plants where evacuation orders remained. Okuma was one of the towns in which the damaged nuclear reactors were located, and the evacuation order remained until 2019. From 2013, residents were allowed to return to Okuma, but only during the daytime. In April 2019, following environmental decontamination endeavors and subsequent low radiation doses, the evacuation order was lifted for specific areas. Even in 2022, the evacuation order has not been fully lifted, and the return rate is low (3.5%, 356 returnees) [9]. During the eight years of evacuation faced by Okuma residents, many families, especially those with young children and working professionals, settled in their evacuated areas and integrated themselves into new communities at their new places of residence.

The acute and long-term impacts of both the Fukushima disaster itself and the associated evacuation process have manifested in the form of psychological distress [10,11], depression [12], PTSD [13], increased rates of obesity [14], kidney dysfunction [15], dyslipidemia [16], hyperuricemia [17], diabetes [18], metabolic syndrome [19], cardiovascular impairment [20,21], musculoskeletal pain [22], bone fractures [23], harmful health behaviors [24,25], and increased suicide rates [26]. Studies have demonstrated that the effects of the disaster have been exacerbated in the elderly [23,27–29]. Enduring such a nuclear disaster and the resulting evacuation leads to worsening of disability, and impaired cognition [30] alongside the losses in social, intellectual, and physical functions that occur as a result of the physiological ageing process [31].

Data on current demographic trends collected by the Okuma Town Office [32] indicate that the proportion of elderly returning to Okuma has been steadily increasing since 2012. A

survey conducted by the Japanese government on Okuma residents found that an interest in returning was expressed by mostly elderly evacuees [33]. Our previous surveys conducted in Tomioka [34] and Kawauchi [35] found that older men who were not living with children were most likely to be interested in returning, and this group had a higher interest in seeking information regarding living in their hometowns, including participating in communication sessions related to risk communication and town recovery efforts. Thus, 11 years after the FDNPP accident, it is essential to survey the quality of life of the elderly and unveil target areas requiring interventions. This study aims to examine the role of age in self-reported quality of life among residents of Okuma town. Specifically, the study focuses on how the perception of risk of returning to reside in Okuma town and thus, the intention to return, differs between elderly and non-elderly residents. As demonstrated in previous studies, there is an association between risk perception, intention to return and quality of life among evacuees. We hypothesized that elderly residents would perceive lower risk than their younger counterparts, and thus express a higher intention to return. Additionally, we predicted a lower self-reported quality of life among the elderly because of prolonged evacuation. We predict that the concern regarding the release of treated water will be equally high among all Okuma residents, regardless of sex, age or any other demographic characteristics or attributes. These results can enable the formation of targeted interventions to improve health status and guide policies towards the effective reconstruction of the affected areas. They may also serve as guidelines for future disaster mitigation and protection for this population group.

## Materials and methods

### Participants

This study was conducted in Okuma town, located within Fukushima prefecture, in January 2022. The study participants were current residents and evacuees aged ≥20 years who held resident cards for Okuma town as of December 2021 and who were able to receive mail from the municipal office. Questionnaires were distributed by the municipal office to most households, with two sets per household. In the instructions provided in the questionnaire, we asked that if the household consisted of one study participant, that the other questionnaire be discarded. If there were three or more target participants in the household, additional questionnaires would be mailed to them. The total population in Okuma was approximately 10,100 (5,000 males and 5,100 females), and the number of total households was approximately 4,800. Excluding residents aged less than 19 years, the distribution of the elderly and non-elderly, according to national statistics for Okuma town was 3500 persons or 34.7% (1,600 or 32.0% elderly males and 1,900 or 37.2% elderly females) and 4,700 persons or 46.5% (2,400 or 48.0% non-elderly males and 2,300 or 45.0% non-elderly females), respectively. The data collection period was from 6 January to 2 March 2022. The basis and purpose of the study were explained in a letter attached to the questionnaire, along with a privacy notice. Written informed consent was obtained from all participants through the return postage of the questionnaire. All study protocols were approved by the ethics committee of Nagasaki University Graduate School of Biomedical Sciences (approval No. 21082702–2, 25 November 2021).

### Questionnaire

The questionnaire used in this study was based on the version used in previous studies conducted within affected towns of Fukushima prefecture [35–39] and in the Fukushima Health Management Survey [40]. The present questionnaire contained questions related to participant demographics (sex, age, family structure, currently living with a child or not, location at the time of the FDNPP accident, current employment, economic lifestyle and interaction with

friends), intention to return to Okuma (thinking of returning, familial disagreements regarding returning, desire to know about life in Okuma, desire to attend events held in Okuma), self-perceived knowledge level regarding radiation risks (understanding of effects of radiation on the human body, desire to learn about the basics of radiation and its effects on the human body, desire to learn about treated water released from the FDNPP), risk perception (anxiety regarding consumption of food harvested in Okuma, drinking tap water from Okuma and self-health effects on future generations). Responses were in the form of yes/no, or where appropriate, as a multiple-choice answer. Questions regarding perception and knowledge were scored using a 4-point scale (1 = strong yes, 2 = probably yes/a lot, 3 = probably no/a little, 4 = strong no).

Quality of life was assessed using the validated [41] Japanese version of the HR-QoL Short Form-8 (SF-8) scale [42], which measures health status on eight dimensions: general health, physical function, physical role (limitations in role due to physical health dysfunction), bodily pain, vitality, social function, mental health, and role emotional (limitations in role due to emotional health dysfunction). Answers were provided on a 5- or 6-point response scale, ranging from 1 (very good/not hindered at all) to 5 or 6 (very bad/inability to function). The SF-8 is interpreted based on scaled scores for two broad classifications: the Physical Component Summary (PCS; comprising general health, physical function, physical role, and bodily pain) and the Mental Component Summary (MCS; comprising vitality, social function, mental health, and role emotional). Scores higher than 50 were considered to indicate good health, based on mean values among the general population in Japan [42].

## Statistical methods

The present study analyzed the role of age on risk perception and quality of life. Participants were divided into two age groups, young (<65 years) and elderly (≥65 years) [40]. The proportions (%) and number (N) of responses were derived for each response category. The factors that played a significant role in risk perception and quality of life with respect to age group were identified using the chi-square test. Quality of life using the SF-8 scale was assessed by age groups using mean (and standard deviation) scores, and also with a cut-off value of 50. Logistic regression analysis was then conducted on the identified variables to assess the independence of their effects on risk perception and quality of life among the elderly. Odds ratios (ORs) with 95% confidence intervals (95%CI) were obtained. Data analysis was performed using IBM SPSS Statistics version 28. p-values <0.05 were considered statistically significant.

## Results

### Demographic characteristics

From the 4,440 households, 940 responses were received. After excluding the 71 responses in which age was missing, 869 were included in the analysis. The response rates for elderly were 15.5% (248 / 1,600) for males and 12.6% (240 / 1,900) for females, and for non-elderly were 7.8% (187 / 2,400) for males and 8.4% (194 / 2,300) for females. Among all respondents, 51.9% were elderly and 40.5% were non-elderly. Among the elderly, most lived either alone (25.5%) or in a 2-person family structure (51.2%) (p < .001) (Table 1). Most elderly residents were not living with a child or grandchild (90.2%) (p < .001), and 97% resided within Fukushima at the time of the accident (p = 0.018). Most of the respondents were not currently employed (89.6%) (p < .001). The age groups did not differ significantly based on sex or current financial status. Almost 60% of the elderly reported that they frequently or regularly interacted with friends or friend groups that had been created before the disaster (p = 0.007).

## Risk perception for radiation exposure, self-perceived knowledge level and desire to know about radiation

Among those reporting anxieties regarding the health effects of radiation on future generations, a higher proportion was from the elderly group (55.8%), than from the younger group

**Table 1. Demographic characteristics, intention to return, anxiety, and desire to know about radiation.**

| Variable | Reference | ≥65 years [a] (n = 488) | <65 years [a] (n = 381) | p-value |
|---|---|---|---|---|
| Sex (n = 869) | Male | 248 (50.8) | 187 (49.1) | 0.633 |
| | Female | 240 (49.2) | 194 (50.9) | |
| Family structure (n = 868) | Living alone | 115 (25.5) | 119 (18.8) | < .001* |
| | 2 persons | 231 (51.2) | 119 (32.4) | |
| | ≥ 3 persons | 105 (23.3) | 179 (48.8) | |
| Living with children/grandchildren (n = 846) | No | 431 (90.2) | 263 (71.5) | < .001* |
| | Yes | 47 (9.8) | 105 (28.5) | |
| Location during accident (n = 834) | Within Fukushima | 456 (97.0) | 340 (93.4) | 0.018* |
| | Outside Fukushima | 14 (3.0) | 24 (6.6) | |
| Employment status (n = 802) | Employed | 47 (10.4) | 212 (60.6) | < .001* |
| | Unemployed | 405 (89.6) | 138 (39.4) | |
| Financial status (n = 845) | Uncomfortable | 115 (24.6) | 103 (27.3) | 0.385 |
| | Comfortable | 353 (75.4) | 274 (72.7) | |
| Anxiety regarding consumption of food produced in Okuma (n = 864) | Yes | 252 (52.2) | 179 (47.0) | 0.132 |
| | No | 231 (47.8) | 202 (53.0) | |
| Anxiety regarding consumption of tap water from Okuma (n = 863) | Yes | 293 (60.5) | 222 (58.6) | 0.576 |
| | No | 191 (39.5) | 157 (41.4) | |
| Anxiety regarding health effects from radiation exposure in Okuma (n = 850) | Yes | 252 (53.6) | 192 (50.5) | 0.407 |
| | No | 218 (46.4) | 188 (49.5) | |
| Anxiety regarding radiation health effects on future generations in Okuma (n = 840) | Yes | 259 (55.8) | 179 (47.6) | 0.018* |
| | No | 205 (44.2) | 197 (52.4) | |
| Self-perceived knowledge level of the effect of radiation on human body (n = 862) | Yes | 266 (55.3) | 240 (63.0) | 0.026* |
| | No | 215 (44.7) | 141 (37.0) | |
| Desire to know about basics of radiation (n = 862) | Yes | 326 (68.5) | 244 (64.9) | 0.272 |
| | No | 150 (31.5) | 132 (35.1) | |
| Desire to know about treated water released from FDNPP (n = 859) | Yes | 371 (76.8) | 269 (71.5) | < .001* |
| | No | 112 (23.2) | 107 (28.4) | |
| Desire to know about life in Okuma (n = 851) | Yes | 394 (83) | 277 (73.6) | 0.003* |
| | No | 81 (17.1) | 99 (26.3) | |
| Desire to attend events in Okuma (n = 851) | Yes | 219 (46.1) | 146 (38.8) | < .001* |
| | No | 256 (53.9) | 230 (61.2) | |
| Intention to return (n = 859) | Already return or Yes | 91 (18.8) | 38 (10.1) | < .001* |
| | Unsure | 117 (24.3) | 125 (33.2) | |
| | No | 274 (56.8) | 214 (56.8) | |
| Familial disagreements regarding intention to return (n = 755) | Yes | 95 (23.2) | 95 (27.5) | 0.179 |
| | No | 315 (76.8) | 250 (72.5) | |
| Interaction with friends (n = 860) | Yes | 289 (59.8) | 188 (49.9) | 0.007* |
| | Can't say | 61 (12.6) | 61 (16.2) | |
| | No | 133 (27.5) | 128 (34.0) | |

[a] N (%).

*p-value <0.05.

(47.6%). There were no significant differences in risk perception regarding the consumption of food or water from Okuma, or regarding effects on self-health, based on age. A lower proportion of elderly residents (55.3%) than younger residents (63.0%) reported self-perceived knowledge regarding the effects of radiation on the human body (p = 0.026). There were no differences between the two age groups regarding the desire to learn the basics of radiation, but there were differences regarding the desire to know more about treated water released from the FDNPP, where 76.8% of elderly residents and 71.5% of younger residents expressed a desire to know more (p<0.001).

## Intention to return

Compared with the young age group, more persons in the elderly age group expressed a wish to return (elderly, 18.8% vs younger residents, 10.1%). A lower proportion of elderly residents reported feeling unsure regarding returning (elderly, 24.3% vs younger residents, 33.2%), but the same proportion in each group decided not to return to Okuma at all (56.8%) (p<0.001). There were no differences according to age in familial disagreements regarding the intention to return.

## Quality of life

Poor physical health was reported by 68.8% of the elderly group (p<0.001) but there were no age-related differences concerning mental health (Table 2). In the individual physical health components, 69.4% reported poor physical function (p<0.001), 69.0% reported poor physical role (p<0.001), 59.7% reported bodily pain (p = 0.015), but only 44.0% reported poor general health (p = 0.016). In the mental health components, 65.1% felt that their social functioning was affected (p<0.001) and 71.1% reported a dysfunctional role emotional component (p = 0.004). Differences with regard to vitality could not be detected.

Odds ratios regarding risk perception and quality of life were obtained for the elderly residents, with the younger residents as the reference group. Logistic regression revealed that the odds ratio for knowing more about FDNPP-treated water (OR 1.3, 95%CI 1.1–1.5, p = 0.001) and odds ratio for having anxiety regarding the health of future generations (OR 1.4, 95%CI 1.0–1.8, p = 0.034) were independently associated with the elderly group (Table 3). In the

**Table 2. Quality of life.**

| Variable | Mean score ± Standard deviation | | Score <50 | | |
|---|---|---|---|---|---|
| | ≥65 years | <65 years | ≥65 years [a] n = 461 | <65 years [a] n = 380 | p-value |
| General health | 48.3 ± 6.8 | 49.3 ± 7.4 | 203 (44.0) | 136 (35.8) | 0.016* |
| Physical function | 46.6 ± 7.2 | 49.2 ± 6.6 | 320 (69.4) | 185 (48.7) | < .001* |
| Physical role | 46.4 ± 7.9 | 49.3 ± 7.1 | 318 (69.0) | 190 (50.0) | < .001* |
| Bodily pain | 46.0 ± 9.0 | 47.6 ± 9.3 | 275 (59.7) | 194 (51.1) | 0.015* |
| Physical component score | 45.5 ± 7.3 | 48.3 ± 6.9 | 317 (68.8) | 200 (52.6) | < .001* |
| Vitality | 49.0 ± 6.0 | 49.5 ± 6.7 | 300 (65.1) | 223 (58.7) | 0.063 |
| Social function | 45.6 ± 8.6 | 47.6 ± 8.5 | 300 (65.1) | 202 (53.2) | < .001* |
| Mental health | 47.7 ± 6.8 | 47.7 ± 7.8 | 238 (51.6) | 187 (49.2) | 0.489 |
| Role emotional | 46.9 ± 7.5 | 47.9 ± 7.8 | 328 (71.1) | 234 (61.6) | 0.004* |
| Mental component score | 47.2 ± 6.9 | 47.0 ± 7.8 | 280 (60.7) | 220 (57.9) | 0.438 |

[a]N (%).

*p-value <0.05.

**Table 3. Logistic regression–risk perception and quality of life for elderly residents.**

|  | Variable | Reference[1] | OR | 95% CI |
|---|---|---|---|---|
| Risk perception | Sex | Male/female | 1.125 | 0.848–1.492 |
|  | Current economic status | Uncomfortable/comfortable | 0.785 | 0.567–1.086 |
|  | Anxiety regarding radiation health effects on future generations | Yes/no | 1.368* | 1.024–1.827 |
|  | Desire to know about FDNPP treated water | Yes/no | 1.295* | 1.109–1.512 |
| Quality of life | Sex | Male/female | 1.297 | 0.971–1.734 |
|  | Current economic status | Uncomfortable/comfortable | 0.893 | 0.745–1.071 |
|  | Physical component score | <50/≥50 | 2.229* | 1.642–3.025 |
|  | Mental component score | <50/≥50 | 1.163 | 0.859–1.574 |
|  | Interaction with friends | Yes/No | 1.337* | 1.165–1.534 |

OR: Odds ratio; CI: Confidence interval

*p-value <0.05.

[1]Reference group is young residents (age<65 years).

elderly group, physical component health summary scores were independently worse (OR 2.2, 95%CI 1.6–2.9, p<0.001) and interaction with friends was significantly better (OR 1.3, 95%CI 1.2–1.5, p<0.001) compared with the younger group (Table 3).

## Discussion

Due to advances in health, and technology, as well as developments in the economic and social sectors, the world is witnessing a revolution in human longevity. The elderly population is predicted to increase from a fifth of the current global population to 61% by 2100 [43]. Japan currently leads the global old-age dependency ratio (OADR) with the elderly comprising 30% of the national population [44]. Those aged 65 years or older are more prone to morbidity and mortality, and these risks are further magnified in disaster settings. Due to reduced functional capacity, the elderly are susceptible both to acute threats to life during the disaster and evacuation process, and to chronic disturbances in mental and physical health resulting from the long-term effects of relocation, progression of chronic diseases, and interruption of access to health services [45].

In this study, conducted 11 years after the FDNPP accident, we investigated the differences in risk perception for radiation exposure and self-reported mental and physical health status between elderly and non-elderly residents of Okuma. Our results show that compared to younger residents, the elderly had a 1.4 times higher odds ratio (95%CI 1.0–1.8, p = 0.034) of having anxiety about radiation-related health effects on future generations and a 1.3 times higher odds ratio (95%CI 1.1–1.5, p = 0.001) of wanting to know more about the release of FDNPP-treated water into the environment. Anxiety is a meaningful emotion experienced by humans as it enables individuals to perceive risk and avoid hazardous situations. Feelings of anxiety and risk perception are also substantially dependent on an individual's past experiences and the pattern of that individual's mental heuristics [46]. As such, general feelings of anxiety are normal and temporary in most individuals. However, after experiencing a disaster, a long-term evacuation and the possible resulting PTSD, such feelings of anxiety might be uncontrollable and persistent, leading to misjudgment of risk and long-term detrimental effects on health and quality of life [47]. There is currently no scientific evidence in humans establishing the increased occurrence of hereditary diseases in offspring from parental radiation exposure [48–51].

It is essential to reiterate and underscore these findings clearly and concisely to the elderly to prevent misunderstandings and inflated risk perceptions. A survey conducted in Fukushima in 2012 found that compared to residents aged 15–49 years, residents ≥65 years felt that immediate health effects and genetic effects were very likely to occur [11]. Our study showed similar results for genetic effects only. Similarly, a study set in the neighboring town of Kawauchi, demonstrated that compared to those < 60 years, those aged >60 had a 1.48 times higher odds ratio for perceiving genetic risk from radiation. Comparable results were found in 2015, 2017 and 2021 [52], which demonstrates the persistence of risk perception in these residents.

A further source of risk perception stems from the Japanese government's recently announced plans to dispose of FDNPP-exposed water into the surrounding sea [53]. Ground-water and rainwater entered broken buildings housing the reactors that were damaged by the hydrogen explosion, resulting in the production of water imbued with high concentrations of radioactive materials. To meet the regulatory standards for discharge, the contaminated water was treated using filtration systems such as the Advanced Liquid Processing System (ALPS) to remove radioactive nuclides. As of 2022, the Ministry of Economy, Trade, and Industry, Japan and the IAEA stated that this was achieved for almost all radionuclides except for tritium [54]. The resulting tritium water is currently being stored in a tank on the FDNPP site and is await-ing disposal to facilitate the process of decommissioning the plant [55]. The IAEA has affirmed that this process is "technically feasible and in line with international practice [56]". However, separately from the concerns of Okuma residents, some environmental groups, marine organi-zations, and neighboring countries have expressed their concerns about the disposal process [57] in terms of probable contamination of the marine food web and eventual health effects at the population level. It is imperative to further study risk perception and specific worries har-bored by affected residents in detail, and also to measure the long-term ecological and eco-nomic impacts of this decision.

The present study showed that compared with the younger group, elderly residents demon-strated a 2.2 times higher odds ratio for having a poor physical component score on their SF-8 scale. All physical components (general health, physical function, physical role and bodily pain) showed scores of less than 50 among the elderly group. However, a similarly significant correlation between age and mental health was not demonstrable by age group, although it should be noted that poor mental health component scores were found in the majority of resi-dents in both the elderly (61%) and young (58%) groups. Previous studies conducted in Fukushima prefecture have established a relationship between intention to return and quality of life. Orita et al. revealed that those evacuees who were "undecided" regarding their intention to return to their hometowns reported worse quality of life, relative to those who had already returned/planned to return and to those who had decided not to return [38]. In our study, 24% of the elderly group and 30% of all residents were undecided regarding their intention to return, and 72.6% of those who wanted to return to Okuma reported poor physical component scores (p = 0.035). Evidence regarding the long-term physical health effects of evacuation has already been established, explained perhaps by the increased stress and the lack of resources available at evacuation sites to maintain a healthy lifestyle [58], both of which are exacerbated in the elderly. Similar results were demonstrated by Borglin et al. in a non-disaster setting in Sweden, where reduced physical mobility was associated with lower self-reported quality of life, and this was more pronounced in elderly women [59]. Our results revealed that interac-tion with friends was significantly better among the elderly than among younger residents. High levels of interactions with friends (a source of social support) were perhaps the reason that this effect did not extend towards a poor mental health component score. This result is congruous with the results of studies from South Korea [60] and Japan [61,62], in which self-

rated good health among the elderly improved with increased social interaction. Therefore, the establishment of access to health services for the elderly is crucial to improve their health conditions and preventing the progression of chronic diseases. Improvement of the quality of the surrounding built environment, which facilitates health-promoting behaviors and peer interactions, can delay the worsening of functional capacity [43] and thus improve the quality of life in the elderly.

Our results highlight several crucial considerations regarding the quality of life of elderly evacuees and residents of Okuma. The probable low risk of transgenerational effects of radiation should be communicated and disseminated to the elderly clearly and concisely. The environmental effects of the disposal of FDNPP-treated water into the surrounding waters should be further examined, and residents' fears regarding the long-term effects of this act must be assessed. It is necessary to undertake health promotion targeted at improving the quality of the surrounding environment of the elderly to facilitate health-promoting behaviors, along with improving access to health services for regular monitoring of chronic diseases. Opportunities to increase social interaction with friends and interactions with experts should be encouraged in earnest, for both elderly and younger residents, to tackle the issue of poor mental health and to improve well-being.

The main limitation of our study was that this was a cross-sectional survey-based report, and thus only correlations, and not causal relationships, could be derived. Due to the nature of the 2011 disaster, various stakeholders, including government officials, academic institutions, and town councils have undertaken surveys of these residents. With time, residents naturally face survey fatigue, especially for voluntary questionnaires, compared with mandatory government-led surveys. A typical response rate for postal questionnaires, such as ours is around 20% [63]. The low response rate of 21% in the present study might have caused respondent bias if residents interested in returning or those who felt an attachment to their hometown were more likely to respond. Although response rates to questionnaires in this area have decreased over time (as demonstrated in the Fukushima Health Management Survey responses, from 40% in 2012 to 20% most recently), our aim in conducting this research was to aid in the rehabilitation of Okuma town, including respecting the wishes of those residents who do not wish to return. A small proportion of elderly residents reported a wish to return, but the majority did not. This is perhaps a message in and of itself, indicating the need to refocus efforts to attract new residents into the town. Nevertheless, the desires of elderly residents who wish to return cannot be disregarded. This information is valuable to the Okuma town council for facilitating the return of residents, especially older people who wish to do so, while also providing insights to the international community regarding the realities of post-nuclear accident rehabilitation. Thus, the present results may represent the opinions of these residents specifically rather than all Okuma residents.

As a population subset, elderly residents are more vulnerable and at higher risk of morbidity and mortality than the younger group, especially during and after disasters. It is essential to support their rehabilitation and improve their quality of life after long-term evacuation. This research can assist Okuma Town and its residents in undertaking reconstruction efforts and in future disaster preparedness. The results of future longitudinal analyses will be an effective tool for monitoring long-term health effects in evacuees and returnees. According to a survey conducted by the central government of Japan in 2021 on Okuma residents' intention to return, approximately 16% had already returned or wanted to return, 23% were undecided, and 57% had decided not to return [33], which is consistent with the results of this study. As the rate of return increases, those who have already returned and evacuees who wish to return to Okuma should be analyzed separately; currently, however, the limited sample sizes would give rise to

biases. Future studies should also examine residents' concerns regarding the disposal of treated water from the FDNPP, as this may affect intentions to return.

## Conclusions

Elderly residents of Okuma town displayed a higher odds ratio for perceived risk regarding the transgenerational effects of radiation and have a higher desire to seek information regarding the disposal of FDNPP-treated water compared with younger residents. The latter implies a perception that health problems could arise in future Okuma-residing residents as a result of accumulated radiation exposure in their parents and the environment. In addition, the elderly displayed lower self-reported physical health compared to younger residents. Clearing misconceptions and disseminating coherent information is key to lowering risk perception among this group. Further in-depth research among residents is required regarding the disposal of FDNPP-treated water and the perceived risks of this action. Health promotion through the encouragement of social participation, improvement of the built environment to facilitate healthy behaviors, and enhanced access to health services are fundamental to improving the quality of life of the elderly population in Okuma.

## Supporting information

**S1 File.**
(XLSX)

## Acknowledgments

We would like to thank all study participants and staff members of the municipal government of Okuma. This study was conducted as a collaboration between Nagasaki University and Okuma Town Council.

## Author Contributions

**Conceptualization:** Varsha Hande, Makiko Orita, Noboru Takamura.

**Data curation:** Varsha Hande.

**Formal analysis:** Varsha Hande.

**Funding acquisition:** Noboru Takamura.

**Investigation:** Makiko Orita, Yuya Kashiwazaki.

**Methodology:** Makiko Orita.

**Project administration:** Noboru Takamura.

**Resources:** Makiko Orita.

**Software:** Makiko Orita.

**Supervision:** Noboru Takamura.

**Validation:** Varsha Hande, Hitomi Matsunaga.

**Visualization:** Makiko Orita.

**Writing – original draft:** Varsha Hande, Makiko Orita.

**Writing – review & editing:** Yasuyuki Taira, Noboru Takamura.

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
