## [Decision Letter · Decision Letter 0]

12 Oct 2022

PONE-D-22-24853The role of age in self-reported quality of life among elderly residents in Okuma town, Japan in a post-disaster settingPLOS ONE

Dear Dr. Orita,

Thank you for submitting your manuscript to PLOS ONE. After careful consideration, we feel that it has merit but does not fully meet PLOS ONE’s publication criteria as it currently stands. Therefore, we invite you to submit a revised version of the manuscript that addresses the points raised during the review process. Concerns were raised about the case study aspect of the work, although this is not really clear from the paper.  Please address these in a revision as well as other concerns raised by the reviewers.

We look forward to receiving your revised manuscript.

Kind regards,

Sakae Kinase, Ph.D.

Academic Editor

PLOS ONE

Journal Requirements:

Reviewers' comments:

Reviewer's Responses to Questions

**Comments to the Author**

1. Is the manuscript technically sound, and do the data support the conclusions?

Reviewer #1: Partly

Reviewer #2: Yes

2. Has the statistical analysis been performed appropriately and rigorously? 

Reviewer #1: Yes

Reviewer #2: Yes

3. Have the authors made all data underlying the findings in their manuscript fully available?

Reviewer #1: Yes

Reviewer #2: Yes

4. Is the manuscript presented in an intelligible fashion and written in standard English?

Reviewer #1: Yes

Reviewer #2: Yes

5. Review Comments to the Author

Reviewer #1: This manuscript purposed to examine the difference in the association among indicators of self-reported quality of life and other socio-psychologic factors, risk perception of radiation exposure including self-perceived knowledge level and desire to know about radiation, intention to return, and self-reported quality of life, comparing the association between the age groups of elderly and younger. The manuscript was written at an average level in the manner, but contains some crucial problems.

1) The title of “The role of age in self-reported quality of life among elderly residents…” seems to imply that the investigated indicators were compared between age classes among the elderly and some specific analysis of the effects of age on those factors, but the contents of manuscript was simply the comparison between elderly and younger groups. So, the title may need to be modified to present the real contents.

2) It looked slightly ambiguous which association was focused among the investigated factors (aspects of investigations) in this study because there were many factors with various aspects. Authors may need to clarify their hypothesis about association rather than simple explore of association.

3) L59-62: The contents of this sentence is incorrect. The concept of radiation restriction for the public by ICRP was not solely by Mettler’s paper, but by substantially wide accumulation of knowledge. The level of radiation exposure for the public in the situation of “existent exposure” such as aftermath of nuclear accident is not “limit” but “reference level” that should adapt the real situation and should be changed in accordance with the efforts for improvement of environment. Authors should learn about those concept by the ICRP Publications 103 and 111. Otherwise this sentence should be deleted.

4) L67: The word “declared” is inappropriate. Either body (WHO or UNCEAR) cannot and did not “declare” it. It might be replaced with “suggested” or some other.

5) L97-100: The response rate to the survey was 21% (940/4400) and was too low. Reviewer thinks that it has been said in general that response rate of questionnaire surveys by municipality offices was around half in Japan. Recipients of the questionnaire might be reluctant with the too long list of questions, academic-oriented questions rather than cuddling up to their situation, or other reasons. Authors should have thought to keep the response rate as low response rate should induce serious selection bias. In addition, the response rate among each of elderly and younger groups needs to be reported.

6) L158-166: It seems slightly curious that the authors asked the residents only about “desire to know more about treated water released form the FDNPP.” It seems necessary to compare it with the knowledge and attitudes of residents to radioactivity pollution on land and activities of countermeasures against it.

Minor points:

1) L42-58: Explanation of the situation at the time of the accident seems to be too detailed although it seemed to be little associated with the contents of this manuscript. This might be shortened.

2) Abstract, L32, 149, 160 (twice), 172, 175, 189, 246: The word “cohort” is inappropriate, but “group” is appropriate in this study design.

3) L217: “the odds” might be “odds ratio.”

Reviewer #2: General comment

This study, which uses a questionnaire survey to understand the situation of residents in difficult-to-return areas affected by the nuclear disaster, is important not only for the development of post-disaster public health activities, but also for the efforts of local government administration.

The conclusion of the paper should briefly state the results of testing the research hypothesis based on the data obtained in the study and its interpretation. For this reason, the conclusions of this paper should be more compact. The content the authors wish to discuss in the conclusions section should be moved to the discussion section.

It would be good to have a discussion on the impact of the measures on the significant differences found in terms of effect size.

Specific comment 1 Financial disclosure

The response could not be true because this research fund states that research proposals shall be evaluated by the Research and Survey Project Evaluation Committee and reflected in the review of research plans, etc.

Specific comment 2 Line 60

“Based on evidence that an increase in cancer incidence and mortality occurs with exposure

60 to radiation doses higher than 100 millisieverts (mSv) [5],”

Although this is a quoted and controversial point, there are studies where radiation risks have been found even for exposures with effective or equivalent doses of less than 100 mSv, and measures are being taken for radon based on epidemiological studies.

Ref.

Proposer D. J. Brenner, Opposer O. G. Raabe, Moderator J. C. McDonald. Is the linear-no-threshold hypothesis appropriate for use in radiation protection? Favouring the proposition. Radiat Prot Dosimetry. 2001;97(3):279-82; discussion 285.

Doss, M., Little, M. P., & Orton, C. G. (2014). Point/Counterpoint: low-dose radiation is beneficial, not harmful. Medical physics, 41(7), 070601.

Specific comment 3 Line 98

“We distributed questionnaires to residents, who were able to receive mail from the municipal office (4,400 postal letters). The data collection period for this study was from 6 January to 2 March 2022. Of the 940 responses received, all were included in the analysis.”

The reviewer believes that a distinction should be made between methods and results. Shouldn't the actual numbers distributed and collected be presented as results, if you generalize the research to be conducted in the affected areas?

Specific comment 4 Line 99

Since this is a research study conducted with the cooperation of the town, the reviewer thought it would be a good idea to clearly state the protocol from an ethical standpoint regarding collaboration with the government and residents, as stated in the acknowledgments.

Specific comment 5 Line 128

“Scores higher than 50 ± 10”

Does the breadth of the criteria depend on some condition?

Specific comment 6 Line 143

To determine respondent bias, it would be helpful to present information on the demographic characteristics of not only those who responded to the survey, but also those who were subjects of this research.

Specific comment 7 Line 164

“There were no differences regarding the desire to learn the basics of radiation, but 76.8% of residents who expressed a desire to know more about treated water released from the FDNPP were elderly (p<.001).”

It would be better to make the wording more explicit such as “There were no differences between these age groups regarding the desire to learn the basics of radiation, but there are differences regarding the desire to know more about treated water released from the FDNPP with 76.8% of elderly residents expressed a desire to know with 71.5% of those under age 65 wanting to know (p<.001).”

Specific comment 8 Line 181

“There were no differences with regard to vitality.”

Since p=0.063, it may be considered that the difference is not detected due to lack of power. If the authors claim that there is no difference, they need to show the result of equivalence testing.

Specific comment 9 Line 186

Although understandable in the overall context, it would be better to clarify that the odds ratios are for each factor in the elderly compared to the under 65 age group. At least, Table 3 should be expressed in a way that it can be understood independently.

Specific comment 10 Line 209

Although, as the authors state, no heritable effects have been clearly found in humans, heritable effects of radiation have been found in mammals. Research on heritable epigenetic effects is also needed in humans.

As another point of contention, is ANXIETY itself a problem? Anxiety should be one of the most important emotions in humans.

Niwa, O. Induced genomic instability in irradiated germ cells and in the offspring; reconciling discrepancies among the human and animal studies. Oncogene 22, 7078–7086 (2003) doi:10.1038/sj.onc.1207037

European Commission, Directorate-General for Energy, Epigenetic effects : potential impact on radiation protection : proceedings of a scientific seminar held in Luxembourg on 8 November 2017, Publications Office, 2019, https://data.europa.eu/doi/10.2833/731758

Specific comment 11 Line 209

“It is essential to reiterate and underscore these findings in a clear and concise manner to the elderly to prevent misunderstandings and inflated risk perceptions.”

Is it a risk communication issue that the facts are not being communicated in a way that is easily understood? Rather, isn't the background issue more important?

Specific comment 12 Line 222

While it is true that the hydrogen explosions damaged the reactor buildings, a large amount of radioactive material was released into the environment prior to the hydrogen explosion, with the largest release coming from Unit 2, which did not experience a hydrogen explosion.

Specific comment 13 Line 225

“This was achieved for all radionuclides, with the exception of tritium.”

I-129 was found to be above the standard concentration in April 2019. Several exceedances of standard concentrations were also confirmed for Tc-99.

Specific comment 14 Line 280

“Okuma; however, the sample sizes in the former group are limited…”

Respondent bias is more important in interpreting the results than the small response rate, which is subject to chance variation.

Specific comment 15 Line 281

“Future studies should also focus on exploring residents' fears regarding disposal of the treated water from the FDNPP.. ”

Are residents really feared? And, if so, is it the root problem of this issue that residents are fearing?

6. PLOS authors have the option to publish the peer review history of their article (what does this mean?). If published, this will include your full peer review and any attached files.

Reviewer #1: No

Reviewer #2: **Yes: **Ichiro Yamaguchi

---

## [Author Response · Author response to Decision Letter 0]

8 Nov 2022

Dear Editor,

We wish to express our thanks to the Editor and reviewers for providing feedback regarding our manuscript. Please find our point-by-point responses to the issues raised by the reviewers in the attached file. We have revised some sections of the text and added more information where appropriate.

Sincerely,

Makiko

---

## [Decision Letter · Decision Letter 1]

25 Nov 2022

PONE-D-22-24853R1

Comparison of quality of life between elderly and non-elderly adult residents in Okuma town, Japan, in a post-disaster setting

PLOS ONE

Dear Dr. Orita,

Thank you for submitting your manuscript to PLOS ONE. After careful consideration, we feel that it has merit but does not fully meet PLOS ONE’s publication criteria as it currently stands. Therefore, we invite you to submit a revised version of the manuscript that addresses the points raised during the review process.

Your paper is much better in its present version. However, I still have some proposals for further improvements to make: You should confirm the definition of “response rate” and refine the limitation of your study in understanding characteristics of respondents, as noted by one reviewer. These additions would significantly strengthen your paper as an analysis of cross-sectional study by itself is less novel to the risk perception literature.

We look forward to receiving your revised manuscript.

Kind regards,

Sakae Kinase, Ph.D.

Academic Editor

PLOS ONE

Reviewers' comments:

Reviewer's Responses to Questions

**Comments to the Author**

1. If the authors have adequately addressed your comments raised in a previous round of review and you feel that this manuscript is now acceptable for publication, you may indicate that here to bypass the “Comments to the Author” section, enter your conflict of interest statement in the “Confidential to Editor” section, and submit your "Accept" recommendation.

Reviewer #1: All comments have been addressed

Reviewer #2: All comments have been addressed

2. Is the manuscript technically sound, and do the data support the conclusions?

Reviewer #1: Partly

Reviewer #2: Yes

3. Has the statistical analysis been performed appropriately and rigorously? 

Reviewer #1: Yes

Reviewer #2: Yes

4. Have the authors made all data underlying the findings in their manuscript fully available?

Reviewer #1: Yes

Reviewer #2: Yes

5. Is the manuscript presented in an intelligible fashion and written in standard English?

Reviewer #1: Yes

Reviewer #2: Yes

6. Review Comments to the Author

Reviewer #1: The manuscript has been significantly improved. But the low response rate to the survey is still the crucial problem of this report. Even if the response rate of around 20% is inevitably as authors said (L312-316), small differential distribution of background factors between elderly and younger groups could eventually turn the results upside down by confounding and/or biases. It is essential that the difference in response rate and characteristics of responders between the two groups should be examined, and it would help understanding the results by those who know the characteristics of the subject population well, for examples, members of the local community. But, the reviewer is still afraid if international readers might be misled or confused.

The following points needs to be considered to revise.

1) L69-71: Figures of this sentence look like that the number of subjects is those fallen in the band while the percentage is for the subjects from zero dose to the upper limit. e.g. 1284 seems to be the subjects with 1 to <2 mSv, but 96.8% seems to be for zero to <2 mSv. This should be noted in the manuscript or revised to the corresponding percentage of the band in order to avoid readers confused.

2) L97-100: The release of treated water to the sea needs to be included in the hypothesis because the question was specific to this survey.

3) L157-159. (Important) Authors seemed to misunderstand the “response rate.” Response rate is “number of respondents / number of target subjects” for each of elderly and younger groups. Authors showed the proportion of elderly and younger groups only among the responders. The number of target subjects might be shown at the top row of Table 1. In addition, as related to the general comment, response rate for each sex needs to be indicated as the number of target subjects by sex should be known.

4) Table 1. The sum of 4 cells of sexes and age groups was 869 (248+240+187+194), which is different from 940 in L157 although all of them were included in the analysis.

5) L216-266: This part looks like one paragraph. It is too long and looks including quite different aspects, so had better be divided into 4 paragraphs: L216-225, L225-241, L241-249, L249-266.

6) L318: “other groups” looks ambiguous. Is it “younger group”? Why the word is plural form?

7) L307-329: This paragraph might be divided at L316, or more.

8) L323: “the central government” should be “the central government of Japan.”

9) L322-325: This sentence needs a citation (reference literature).

10) L327: Limited sample size would give rise to some selection bias in responders, but it would not give rise to biases promptly in the results of comparison when background of the compared groups was not differential. However, it is difficult to show it as we don’t have the background information of non-responders.

11) L361, L364: Publication number (111 and 103 for each) would be helpful for readers.

12) Reference: Style/format need to be checked thoroughly.

Reviewer #2: Specific comment 1 Financial disclosure

The response could not be true because this research fund states that research proposals shall be evaluated by the Research and Survey Project Evaluation Committee and reflected in the review of research plans, etc.

Authors’ response

This work was supported as a Research Project on the Health Effects of Radiation, organized by the Ministry for the Environment, Japan. The funders had no role in study design, data collection and analysis, decision to publish, or preparation of the manuscript. After consulting with the funders, we were informed that this description is sufficient.

Additional comment

Since the evaluation committee shall evaluate the study design, and the evaluation shall be reflected in the revision of the study design, if the evaluation committee had made any contribution to this study, it should be clearly stated.

https://www.env.go.jp/chemi/rhm/study/R3_01.html

Specific comment 2 Line 60 of the original manuscript

“Based on evidence that an increase in cancer incidence and mortality occurs with exposure to radiation doses higher than 100 millisieverts (mSv) [5],”

Although this is a quoted and controversial point, there are studies where radiation risks have been found even for exposures with effective or equivalent doses of less than 100 mSv, and measures are being taken for radon based on epidemiological studies.

Excerpts from the authors' responses

For the public under normal conditions, the ICRP recommends limiting annual exposure to a radiation dose below 1 mSv (the first band of 1 mSv or less). The second band (1-20 mSv) is applicable to the protection of persons residing in contaminated areas. Actions should be taken to limit individual annual effective residual dose towards the lower range of the band, with periodic reductions over time.

Additional comment

As Reviewer 1 stated, reference levels rather than dose limits are applied in the existing exposure situation, so it is better to use different terminology.

7. PLOS authors have the option to publish the peer review history of their article (what does this mean?). If published, this will include your full peer review and any attached files.

Reviewer #1: No

Reviewer #2: No

---

## [Author Response · Author response to Decision Letter 1]

9 Dec 2022

We thank both reviewers for their time and effort in providing feedback. We have addressed all the points raised, to the best of our abilities. Please find our point-by-point responses and manuscript revisions as attached.

---

## [Decision Letter · Decision Letter 2]

20 Dec 2022

PONE-D-22-24853R2Comparison of quality of life between elderly and non-elderly adult residents in Okuma town, Japan, in a post-disaster settingPLOS ONE

Dear Dr. Orita,

Thank you for submitting your manuscript to PLOS ONE. After careful consideration, we feel that it has merit but does not fully meet PLOS ONE’s publication criteria as it currently stands. Therefore, we invite you to submit a revised version of the manuscript that addresses the points raised during the review process. There seems to be ambiguous descriptions, particularly in descriptions on your questionnaire survey. 

We look forward to receiving your revised manuscript.

Kind regards,

Sakae Kinase, Ph.D.

Academic Editor

PLOS ONE

Reviewers' comments:

Reviewer's Responses to Questions

**Comments to the Author**

1. If the authors have adequately addressed your comments raised in a previous round of review and you feel that this manuscript is now acceptable for publication, you may indicate that here to bypass the “Comments to the Author” section, enter your conflict of interest statement in the “Confidential to Editor” section, and submit your "Accept" recommendation.

Reviewer #1: All comments have been addressed

Reviewer #2: All comments have been addressed

2. Is the manuscript technically sound, and do the data support the conclusions?

Reviewer #1: Partly

Reviewer #2: Yes

3. Has the statistical analysis been performed appropriately and rigorously? 

Reviewer #1: Yes

Reviewer #2: Yes

4. Have the authors made all data underlying the findings in their manuscript fully available?

Reviewer #1: Yes

Reviewer #2: Yes

5. Is the manuscript presented in an intelligible fashion and written in standard English?

Reviewer #1: Yes

Reviewer #2: Yes

6. Review Comments to the Author

Reviewer #1: The manuscript is continuously improved. But, the followings are still crucial problems.

1) The 1st point that authors replied [Towns that lifted their evacuation orders early (2012)….. the realities of post-nuclear accident rehabilitation] needs to be explained not only to reviewer but to readers. This reviewer could not think that the contents are incorporated sufficiently in the manuscript.

2) Methods of mailing of questionnaires are still ambiguous. It does not matter whether the questionnaires are sent individually or per household, as long as the questionnaires are distributed for all household members aged 20 years or older (adults). If the questionnaires sent to a household are insufficient for the number of people in that household, that is a big problem. A clear statement on this point is needed on the manuscript. Not only in reply to reviewers.

3) Authors need to show the number of residents with age of 20 to 64 years and 65 years or older by sex (approximate numbers would be accepted). Then the (rough) response rates need to be shown specifically for the numbers of respondents by sex and age class (248, 187, 240, and 194 in Table 1).

4) L102: “among all Okuma residents” is ambiguous. Need to say “between sexes, ages, or other characteristics/attributes.”

Minor point: the first letter of “publications 103, 111” needs to be capitalized.

Reviewer #2: (No Response)

7. PLOS authors have the option to publish the peer review history of their article (what does this mean?). If published, this will include your full peer review and any attached files.

Reviewer #1: No

Reviewer #2: **Yes: **YAMAGUCHI Ichiro

---

## [Author Response · Author response to Decision Letter 2]

14 Jan 2023

We thank the reviewers for their time and effort in providing feedback. We have addressed all the points raised to the best of our ability. Please find our point-by-point responses and manuscript revisions as attached.

---

## [Decision Letter · Decision Letter 3]

30 Jan 2023

Comparison of quality of life between elderly and non-elderly adult residents in Okuma town, Japan, in a post-disaster setting

PONE-D-22-24853R3

Dear Dr. Orita,

We’re pleased to inform you that your manuscript has been judged scientifically suitable for publication and will be formally accepted for publication once it meets all outstanding technical requirements.

Kind regards,

Sakae Kinase, Ph.D.

Academic Editor

PLOS ONE

Additional Editor Comments (optional):

I have much pleasure in recommending this paper for publication. The manuscript has been substantially with changes highlighted point by point according to reviewers' comments.

Reviewers' comments:

Reviewer's Responses to Questions

**Comments to the Author**

1. If the authors have adequately addressed your comments raised in a previous round of review and you feel that this manuscript is now acceptable for publication, you may indicate that here to bypass the “Comments to the Author” section, enter your conflict of interest statement in the “Confidential to Editor” section, and submit your "Accept" recommendation.

Reviewer #1: All comments have been addressed

Reviewer #2: (No Response)

2. Is the manuscript technically sound, and do the data support the conclusions?

Reviewer #1: Yes

Reviewer #2: Yes

3. Has the statistical analysis been performed appropriately and rigorously? 

Reviewer #1: Yes

Reviewer #2: Yes

4. Have the authors made all data underlying the findings in their manuscript fully available?

Reviewer #1: Yes

Reviewer #2: Yes

5. Is the manuscript presented in an intelligible fashion and written in standard English?

Reviewer #1: Yes

Reviewer #2: Yes

6. Review Comments to the Author

Reviewer #1: The manuscript would be satisfactory from the viewpoint of description on methodologies and results.

Reviewer #2: (No Response)

7. PLOS authors have the option to publish the peer review history of their article (what does this mean?). If published, this will include your full peer review and any attached files.

Reviewer #1: No

Reviewer #2: **Yes: **Ichiro YAMAGUCHI

---

## [Editor Report · Acceptance letter]

5 Feb 2023

PONE-D-22-24853R3 

Comparison of quality of life between elderly and non-elderly adult residents in Okuma town, Japan, in a post-disaster setting 

Dear Dr. Orita:

I'm pleased to inform you that your manuscript has been deemed suitable for publication in PLOS ONE. Congratulations! Your manuscript is now with our production department. 

Kind regards, 

on behalf of

Professor Sakae Kinase 

Academic Editor

PLOS ONE